# Dietary Flavonoid Intake and Anemia Risk in Children and Adolescents: Insights from National Health and Nutrition Examination Survey

**DOI:** 10.3390/antiox14040395

**Published:** 2025-03-27

**Authors:** Linfeng Li, Zhongwang Wang, Zhengyu Yu, Ting Niu

**Affiliations:** 1Department of Hematology, West China Hospital, Sichuan University, Chengdu 610017, China; linfeng_li@wchscu.edu.cn (L.L.); wangzhongwang@scu.edu.cn (Z.W.); 2State Key Laboratory of Biotherapy, Collaborative Innovation Center of Biotherapy, West China Hospital, Sichuan University, Chengdu 610017, China; 3National Facility for Translational Medicine (Sichuan), West China Hospital, Sichuan University, Chengdu 610017, China

**Keywords:** flavonoids, flavan-3-ols, anemia, oxidative stress, NHANES

## Abstract

Anemia, a global health concern, significantly impacts children and adolescents, impairing their physical and cognitive development. While nutritional deficiencies are primary contributors, oxidative stress has emerged as a key factor in anemia pathogenesis. Flavonoids, known for their antioxidant properties, may play a protective role, but their relationship with anemia in pediatric populations remains underexplored. Using data from the National Health and Nutrition Examination Survey (NHANES) cycles (2007–2008, 2009–2010, and 2017–2018), we analyzed 6815 participants aged ≤20 years to investigate the association between dietary flavonoid intake and anemia risk. Flavonoid intake was assessed via two 24 h dietary recalls, and anemia was defined using WHO hemoglobin thresholds. Multivariable logistic regression and restricted cubic spline (RCS) models were employed, adjusting for sociodemographic, dietary, and lifestyle factors. The results showed that lower dietary flavonoid intake was significantly associated with increased anemia risk. High-intake groups of total flavonoids and flavan-3-ols were linked to a reduced anemia prevalence compared to low-intake groups, with odds ratios (ORs) of OR = 0.641 (95% CI: 0.439, 0.935) and 0.612 (95% CI: 0.406, 0.921), respectively. This study highlights the potential protective role of dietary flavonoids, particularly flavan-3-ols, in reducing the anemia risk among children and adolescents, underscoring the importance of flavonoid-rich diets in anemia prevention.

## 1. Introduction

Anemia, defined by decreased hemoglobin levels, is a major global health issue affecting both developed and developing countries, particularly among children and adolescent [1,2]. Anemia can lead to impaired physical and cognitive development, reduced immune function, and increased overall morbidity and mortality in this age group [3,4]. Although there are various factors that influence the pathophysiology of anemia, nutritional deficiencies, especially of iron, folate, and vitamin B12, are the primary causes of anemia [5]. However, recent evidence has pointed out that reactive oxygen species (ROS)-induced oxidative stress of erythrocytes is another principal factor of anemia.

Flavonoids, as the most well-known antioxidants found in diets, are a diverse group of polyphenolic compounds found in many plant-based foods, including tea, fruits, vegetables, and cocoa [6]. In the US Department of Agriculture’s Food and Nutrient Database for Dietary Studies (FNDD), the dietary flavonoids are divided into six major subclasses: anthocyanidins, flavan-3-ols, flavonols, flavanones, flavones, and isoflavones, each with its distinct bioactivity [7,8,9]. Considering that oxidant damage plays an important role in the production and destruction of erythrocytes, increasing research suggests that flavonoids have a potential protective effect on red blood cells, such as by diminishing free radicals, having a protective effect against membrane lipid peroxidative damage, and improving the survival of red blood cells [10,11,12]. This evidence demonstrates the benefit of intaking flavonoids to reduce the risk of oxidative stress-mediated anemia.

Despite the growing body of evidence on the health benefits of flavonoids, limited research has specifically addressed their role in anemia, particularly in the context of children and adolescents who are vulnerable to nutritional deficiencies. NHANES provides a unique opportunity to explore the relationship between flavonoid intake and anemia risk in a large, diverse sample of the U.S. population.

Our study aims to fill the gap in knowledge by investigating the association between dietary flavonoid intake and the likelihood of anemia in children and adolescents, Given the significant differences in iron metabolism between genders, we also focus on the differential effects between males and females. Additionally, we aim to examine the role of specific flavonoid subclasses in modulating this risk. Using data from the NHANES, we explore these associations and provide insights into potential dietary strategies to mitigate anemia risk, particularly in at-risk subgroups.

## 2. Materials and Methods

### 2.1. Study Population

The NHANES, a nationwide research initiative conducted every two years since 1999, aims to assess the health and nutritional status of the US population. It employs a complex, multistage, probabilistic sampling method, gathering data from a representative sample of approximately 5000 individuals annually. This study used data from three NHANES cycles: 2007–2008, 2009–2010, and 2017–2018. Of the 29,940 participants across these cycles, 12,487 were under the age of 20, and 6815 individuals with information on their dietary flavonoid intake and available blood test data were included in the final analysis. Figure 1 provides detailed information on the sample selection.

### 2.2. Outcome Ascertainment

The outcome of our research was anemia. In this study, anemia was defined in accordance with the diagnostic criteria established by the World Health Organization (WHO) [13]. Specifically, anemia was determined based on hemoglobin concentrations below the established thresholds: for children aged 6 to 59 months, anemia was defined as a hemoglobin level lower than 110 g/L; for children aged 5 to 11 years, the cutoff was set at 115 g/L; and for children aged 12 to 14 years, anemia was diagnosed when hemoglobin levels fell below 120 g/L. In non-pregnant women aged 15 years and older, a hemoglobin level below 120 g/L was considered indicative of anemia, while for pregnant women, the threshold was set at 110 g/L. For men aged 15 years and older, anemia was defined as a hemoglobin concentration below 130 g/L. These hemoglobin cutoffs were employed to categorize participants as anemic for the purposes of this study.

### 2.3. Assessment of Flavonoids

The NHANES distributes dietary survey questionnaires to participants’ households to investigate food intake over the past 24 h. A second 24 h dietary recall questionnaire is conducted via telephone 3 to 10 days later. The specific questionnaire list can be found on the NHANES website (https://wwwn.cdc.gov/nchs/nhanes/search/variablelist.aspx?Component=Dietary&Cycle=2017-2020, accessed on 5 January 2025). To ensure greater reliability and consistency in the research outcomes, this study computed the average dietary intake from the two separate days of questionnaire responses provided by the participants. The FNDD furnishes data on the content of six major subclasses and 29 specific flavonoid compounds across various foods (https://www.ars.usda.gov/northeast-area/beltsville-md-bhnrc/beltsville-human-nutrition-research-center/food-surveys-research-group/docs/fndds-flavonoid-database/, accessed on 5 January 2025). Leveraging the FNDD data, this study quantified the intake of flavonoid compounds from the dietary surveys and calculated the daily consumption of the six flavonoid subclasses—anthocyanidins, flavan-3-ols, flavonols, flavanones, flavones, and isoflavones—among the surveyed population [14]. The total flavonoid intake was determined as the sum of the 29 individual flavonoids. The detailed estimation process was described in a previous study [15].

### 2.4. Covariates

Social demography and lifestyle factors were collected by interviews and questionnaires.

The sociodemographic factors included age, sex (male and female), and race (non-Hispanic White, non-Hispanic Black, and other races). Poverty status was assessed using the Poverty Income Ratio (PIR) and categorized into three groups: low-income (PIR < 1.35), middle-income (1.35 ≤ PIR < 3.0), and high-income (PIR ≥ 3.0) [16]. Individuals were also grouped into normal weight (BMI 18.5–24.9), overweight (BMI 25.0–29.9), and obese (BMI ≥ 30.0) categories based on their BMI. The total daily energy intake, vitamins, iron, and folate intake were obtained from dietary data.

### 2.5. Statistical Analysis

All statistical analyses were performed using R software (version 4.4.1). To ensure accurate estimates that accounted for the complex survey design of NHANES, including the stratification, clustering, and oversampling of specific subgroups, we applied the Mobile Examination Center (MEC) exam weights provided by NHANES. This adjustment also accounted for design changes across different survey cycles. By incorporating the major sampling units, sample weights, and strata, we followed the guidelines recommended by the National Center for Health Statistics to generate nationally representative estimates and weighted analyses which were conducted using the “survey” package in R. Continuous baseline variables were summarized using the median and interquartile ranges (IQR) and compared using Mann–Whitney U tests, while categorical variables were presented as percentages and compared using Chi-square tests. Furthermore, the concentration and distribution of various flavonoid types were analyzed. Statistical significance was considered at a threshold of *p* < 0.05.

Except isoflavones, all other flavonoid compounds were categorized into low, medium, and high groups by evenly distributing the population based on their intake levels. Since 43.79% of the study population had an isoflavone intake of zero, it was not feasible to evenly divide isoflavones into three groups based on intake levels. Consequently, individuals with zero intake were assigned to the low-level group, while the remaining population was equally divided into the medium-level and high-level intake groups according to their consumption.

Multiple logistic regression was employed to examine the association between flavonoid intake and the occurrence of anemia. In the multivariate logistic regression models, Model 1 was adjusted for age and sex; Model 2 was additionally adjusted for ethnicity, family poverty income ratio, energy intake levels (in tertiles), body mass index (normal weight, overweight, or obese); and Model 3 was further adjusted for vitamin B12, vitamin B6, vitamin C, vitamin E, folate, and iron intake levels (in tertiles). To visualize the continuous, non-linear relationship, the links between flavonoid intake and the prevalence of anemia were analyzed using RCS regression with the 10th, 50th, and 90th percentile as nodes and were multivariable-adjusted. Given the skewed distribution of flavonoid compound intake among the population, a logarithmic transformation was applied. A two-sided *p* value of 0.05 was defined as being statistically significant. Likelihood ratio tests were used to test for the interaction between subgroups.

## 3. Results

### 3.1. Participant Characteristics at Baseline

Table 1 presents the survey-weighted sociodemographic and health-related characteristics of 6815 children and adolescents from the NHANES 2007–2010 and 2017–2018 cohorts, stratified by anemia status (310 anemic vs. 6505 non-anemic). Key demographic factors include sex (49% female overall, with 74.06% of anemic participants being female, *p* < 0.001) and ethnicity (55.84% White, 13.45% Black, and 30.71% Other, with Black individuals overrepresented among the anemic group at 39.67%, *p* < 0.001), alongside age and PIR. Dietary intake relevant to anemia includes energy, iron, folate, and vitamins B6 and B12—nutrients critical for erythropoiesis—with significant differences (*p* < 0.05) between groups. Antioxidant intake includes vitamins C and E, which exhibit no significant differences (*p* = 0.71 and *p* = 0.36, respectively), and, notably, the total sum of the 29 flavonoid compounds, which demonstrates a statistically significant difference (*p* = 0.03) between anemic and non-anemic participants.

### 3.2. Distributions and Concentrations of Dietary Flavonoid Intake

Appendix A summarizes the distributions and concentrations of dietary flavonoid intakes. Notably, flavan-3-ols exhibit the highest mean intake (52.22 mg/day), driven largely by thearubigins (23.862 mg/day), followed by flavanones (13.498 mg/day) and flavonols (8.575 mg/day), while isoflavones show the lowest mean intake (1.058 mg/day), with a median of 0.005 mg/day reflecting a highly skewed distribution due to the frequent intake of zero. Anthocyanidins and flavones also display substantial variability, with 95th percentile values reaching 35.473 mg/day and 1.482 mg/day, respectively. The total sum of all 29 flavonoids averages 83.691 mg/day, with a wide range (5.233 to 315.81 mg/day) across percentiles, underscoring the heterogeneity in flavonoid consumption.

### 3.3. Association Between Dietary Flavonoid Intake and the Prevalence of Anemia in U.S. Children and Adolesents

The results of the multiple logistic regression analysis examining the relationship between dietary flavonoid intake and the prevalence of anemia are presented in Table 2. Flavonoid subclasses—isoflavones, anthocyanidins, flavan-3-ols, flavanones, flavones, flavonols, and total flavonoids—are analyzed using the low-intake group as the reference category. The fully adjusted Model 3, which controls for age (1–6, 7–12, 13–20 years), sex (male, female), ethnicity (White, Black, Other), PIR (≤1.35, 1.36–3.0, >3.0), energy intake (tertiles), BMI (normal, overweight, obese), and dietary intake of vitamins B12, B6, C, E, folate, and iron (tertiles), highlights significant protective effects. Flavan-3-ols consistently show a significant inverse association with anemia, with the high-intake group in Model 3 yielding an OR of 0.612 (95% CI: 0.406, 0.921) and a *p*-trend of 0.023. Total flavonoids also exhibit a significant protective effect in Model 3 (OR = 0.641, 95% CI: 0.439, 0.935; *p*-trend = 0.044). Anthocyanidins show significance in Crude and Model 1 (OR = 0.626, 95% CI: 0.428, 0.917; *p*-trend = 0.026), though this is attenuated in later models. Isoflavones, flavones, and flavonols show no consistent trends (*p*-trend > 0.05). Complementing this categorical approach, we further employ RCS curves with the same Model 3 adjustments to visualize the continuous, non-linear relationship between flavonoid intake and anemia risk. The result reveals that total flavonoids (Figure 2A) and flavan-3-ols (Figure 2B) exhibit a clear downward trend in anemia odds as intake rises, with a statistically significant *p* value (*p* for overall < 0.05), suggesting a dose-dependent protective effect that strengthens at higher consumption levels, and this protective effect does not vary across intake levels (*p* for non-linear > 0.05). By integrating tertile-based multiple logistic regression analysis and RCS curves, these results consistently underscore the protective roles of flavan-3-ols and total flavonoids against anemia, offering both statistical robustness and a nuanced view of intake–response dynamics.

### 3.4. Subgroup Analysis of Participants Based on Dietary Total Flavonoids and Flavan-3-Ol Intake Levels

This study conducted subgroup analyses of participants with anemia focusing on their dietary total flavonoids (Table 3) and flavan-3-ols (Table 4). The subgroups are defined by age, sex, ethnicity, PIR, and BMI, with odds ratios (ORs) and 95% CIs adjusted per Model 3 (including age, sex, ethnicity, PIR, energy intake, BMI, and nutrient intake). The P for the trend tests the trend across tertiles within each subgroup, while the P for the interaction assesses whether the association varies across subgroup categories.

For total flavonoids (Table 3), an inverse association with anemia is consistent across most subgroups, with stronger effects in females (medium intake OR = 0.475, 95% CI: 0.297, 0.760; high intake OR = 0.668, 95% CI: 0.475, 0.939; *p*-trend = 0.025) and the ‘Other’ ethnicity group (high intake OR = 0.436, 95% CI: 0.266, 0.714; *p*-trend = 0.002). A significant sex interaction (P for interaction = 0.002) highlights a more robust protective effect in females than males (*p*-trend = 0.113). Other subgroups, like PIR 1.36–3.0 (high intake OR = 0.374, 95% CI: 0.168, 0.835; *p*-trend = 0.019) and obese BMI (high intake OR = 0.366, 95% CI: 0.161, 0.835; *p*-trend = 0.016), also show significant trends, though their interactions are non-significant.

For flavan-3-ols (Table 4), the inverse relationship persists across subgroups, with significant trends in females (high intake OR = 0.555, 95% CI: 0.390, 0.789; *p*-trend = 0.01), the ‘Other’ ethnicity (high intake OR = 0.397, 95% CI: 0.224, 0.700; *p*-trend = 0.008), PIR 1.36–3.0 (high intake OR = 0.347, 95% CI: 0.166, 0.727; *p*-trend = 0.018), and normal weight BMI (high intake OR = 0.544, 95% CI: 0.356, 0.831; *p*-trend = 0.019). However, no significant interactions are observed, suggesting that the association is not strongly modified by these factors. Compared to Table 3 (OR = 0.610, *p*-trend = 0.02), flavan-3-ols show fewer significant subgroup trends.

These results indicate that total flavonoids exhibit a consistent, sex-modified protective effect against anemia, while flavan-3-ols show a steady but less variable association across subgroups, addressing the broader significance seen earlier.

Odds ratios and 95% confidence intervals (CIs) were obtained from multiple logistic regression with restricted cubic splines. The RCS model was adjusted for age (1–6, 7–12, or 13–20), sex (male or female), ethnicity (White American, Non-Hispanic Black, or Other), family poverty income ratio (≤1.35, 1.36–3.0, or >3.0), energy intake levels (in tertiles), body mass index (normal weight, overweight, or obese), and vitamin B12, vitamin B6, vitamin C, vitamin E, folate, and iron intake levels (in tertiles). The median values of each flavonoid’s parameters were chosen as a reference. The panels were as follows: (A) total flavonoids, (B) flavan-3-ols, (C) flavonols, (D) flavone, (E) anthocyanidins, (F) isoflavones, and (G) flavaones. The solid line represented the hazard ratio and the dotted lines the 95% CIs. Areas of gray represented the distribution of the levels of each flavonoid parameter. The P overall was derived by applying a natural logarithm transformation to each flavonoid’s intake and then evaluating it as a continuous variable using multivariable logistic regression. The *p* value for non-linearity was determined using the Wald Chi-square test for linearity.

## 4. Discussion

To our knowledge, this is the first study to demonstrate that lower dietary flavonoid intake is associated with an increased risk of anemia in children and adolescents. Our study first investigated the relationship between dietary flavonoid intake and the likelihood of anemia among children and adolescents, using a nationally representative sample from NHANES. Furthermore, when analyzing flavan-3-ols, the most common flavonoid consumed in the USA, we found no significant interactions with subgroup variables, indicating that the observed relationship was stable and not influenced by other factors.

One possible reason flavonoids may reduce anemia risk is their antioxidative properties, which neutralize free radicals, chelate metal ions, regulate antioxidant/oxidant enzyme activity, and reduce oxidative signaling pathways [17,18]. Oxidative stress, which involves the imbalance between ROS and antioxidant defenses, has been identified as a major contributor to anemia. ROS can damage red blood cells by promoting lipid peroxidation and the breakdown of cellular membranes [19], leading to the increased removal of damaged red blood cells [20,21]. Additionally, increased ROS production activates inflammatory signaling, raising IL-6, TNF-α, and IL-1β levels in the liver. These cytokines upregulate hepcidin expression, reducing iron absorption and inhibiting the release of stored iron from macrophages and liver cells, leading to iron sequestration and functional iron deficiency [22,23]. This effect is particularly pronounced in conditions where oxidative stress is chronic or excessive, such as in certain inflammatory diseases or genetic disorders like sickle cell anemia and thalassemia.

Our research focused on children and adolescents, who are more vulnerable to oxidative stress than adults. Increased energy demands and elevated metabolic rates during puberty can lead to the heightened production of ROS [24,25]. Additionally, adolescents often have poor dietary habits, including an inadequate intake of antioxidants, which are crucial for neutralizing ROS [26,27,28]. As their antioxidant defense systems are still maturing, adolescents are more sensitive to oxidative damage [29]. This makes them particularly prone to oxidative stress-induced anemia, which can impair physical and cognitive development.

The observed gender differences in the association between flavonoid intake and the risk of anemia warrant further investigation. In our study, covariate adjustments have been made for social demographics, lifestyle factors, and major hematopoietic nutrients in the diet, leading us to speculate that these differences are due to endogenous sex-specific factors. Among these, the most noteworthy is the difference in antioxidant mechanisms between males and females. In general, some studies have confirmed that males exhibit a higher production of ROS and less efficient antioxidant mechanisms across several species, including humans [30,31,32], and this seems to be even more pronounced during puberty in humans and animals [33]. This may be attributed to the antioxidant properties of estrogen, gender differences in NADPH oxidase activity, or other underlying mechanisms [34]. Therefore, males may benefit more from the exogenous intake of antioxidants. In addition, due to the effects of menstruation, young women often experience iron deficiency anemia as a result of a negative iron balance. Flavonoids form chelates with dietary iron in the gastrointestinal tract, thereby inhibiting iron absorption, which may impair the protective effects of flavonoids against anemia [35]. These gender-based differences highlight the importance of considering sex-specific factors when studying the impact of diet on anemia.

Our RCS regression analysis revealed that the intake of all types of flavonoids was not non-linearly associated with anemia prevalence in this population. Instead, the data suggest a trend where higher flavonoid intake is linked to a lower risk of anemia. Unlike some prior studies that have indicated a dual effect—where moderate flavonoid consumption might reduce oxidative stress and enhance iron bioavailability, while excessive intake, particularly from sources like tea, could inhibit iron absorption by forming insoluble complexes with non-heme iron—our findings did not show an increased anemia risk in the high flavonoid intake group [36]. This suggests that, within the range of intake observed in this cohort, excessive consumption did not appear to negatively impact iron absorption sufficiently to elevate anemia risk. These results align with emerging evidence that the relationship between flavonoids and anemia may be context-specific, potentially influenced by dietary patterns or baseline iron status [37]. Further studies are warranted to confirm these observations and explore the thresholds of flavonoid intake that optimize anemia prevention without adverse effects on iron metabolism.

The consistency of our findings across different sociodemographic groups was another important aspect of our study. The lack of significant interactions with other variables suggested that the association between flavonoid intake and anemia risk was stable across these subgroups. This is particularly noteworthy because many other dietary factors, such as overall intake of total energy, vitamin B6, vitamin B12, folate, and iron, can vary significantly across different population groups [5]. The robustness of the relationship between flavonoids and anemia indicated that flavonoid-rich diets may be beneficial in reducing anemia risk in a wide range of adolescents, regardless of their background.

Our study also highlighted the importance of considering the role of specific flavonoid subclasses in anemia prevention. Flavan-3-ols, the most common flavonoid consumed in the American adolescent diet, were the focus of our analysis, as they are particularly abundant in commonly consumed foods such as tea, fruits, and cocoa [38]. Flavan-3-ols did not show any significant interactions with subgroup variables in our study. This suggests that the relationship between flavan-3-ol intake and anemia risk is stable across different population groups, further supporting the notion that flavonoids, in general, may have a protective effect against anemia, particularly in male adolescents. Flavan-3-ols have been shown to behave as antioxidants via several mechanisms, similar to other flavonoids [39]. They have been confirmed to be superior to flavonols, the second most consumed flavonoids in the American adolescent diet, in their antioxidant capacity because the oxidation of flavan-3-ols produces semiquinone radicals that couple to produce oligomeric compounds through nucleophilic addition [40]. Additionally, the lower intake of other subclasses of flavonoids may have limited our significance analysis. Future research should explore the role of other flavonoid subclasses and their potential synergistic effects with other nutrients in preventing anemia.

Despite the promising results, our study has several limitations that must be considered. First, the cross-sectional design of NHANES limited our ability to establish causality between flavonoid intake and anemia risk. While the association between reduced flavonoid intake and increased anemia risk is compelling, longitudinal studies are needed to confirm whether flavonoids can directly reduce the risk of developing anemia over time. Second, while we averaged two 24 h dietary recalls to enhance accuracy, this approach may still not have fully captured habitual flavonoid intake compared to longer term methods like food frequency questionnaires or 72 h recalls including weekend days, potentially introducing recall bias or missing typical consumption patterns. Although NHANES uses validated food frequency questionnaires, the accuracy of self-reported data can vary, especially among adolescents. Third, although flavonoids have been shown to inhibit iron absorption in some contexts [41], our study lacks serum iron data to assess this effect, and our results primarily reflect associations with hemoglobin-defined anemia linked to oxidative stress rather than iron status. Lastly, our analysis did not account for other potential confounders, such as underlying medical conditions or genetic anemias, that could influence both flavonoid intake and anemia risk. Future studies should address these limitations by using more robust dietary assessment methods and incorporating a wider range of potential confounding factors, especially among adolescents with sickle cell disease and thalassemia.

## 5. Conclusions

In conclusion, our study suggests that reduced flavonoid intake is linked to an increased risk of anemia, particularly among male adolescents. This highlights the potential role of flavonoid-rich diets in anemia prevention, likely through their antioxidant effects and impact on iron metabolism. However, more research is needed to understand how flavonoids influence anemia risk, especially in females and across different flavonoid subclasses. Longitudinal studies and clinical trials are necessary to confirm these findings and explore the potential for flavonoid-based interventions in at-risk populations.

## Figures and Tables

**Figure 1 antioxidants-14-00395-f001:**
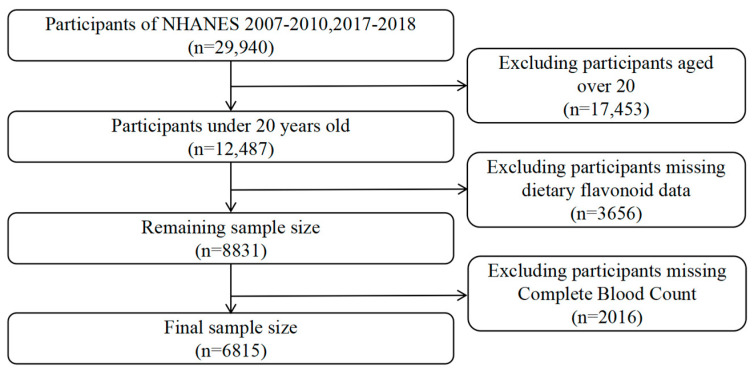
Flowchart of the study design and participants.

**Figure 2 antioxidants-14-00395-f002:**
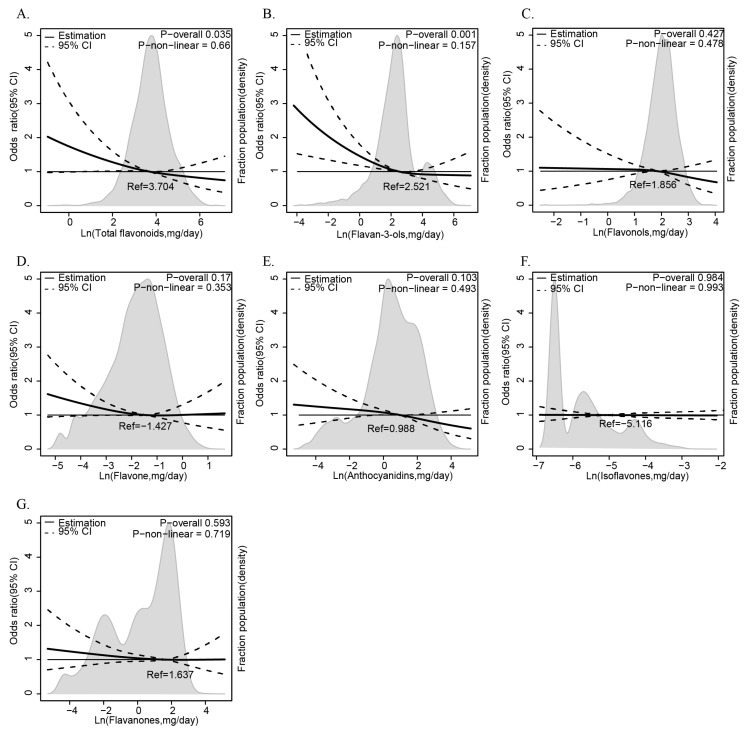
The relationship between dietary flavonoid intake levels (mg/day) and anemia among children and adolescents in NHANES 2007–2010 and 2017–2018. The relationship between flavonoids ((**A**) total flavonoids, (**B**) flavan-3-ols, (**C**) flavonols, (**D**) flavone, (**E**) anthocyanidins, (**F**) isoflavones, (**G**) flavaones) intake and anemia risks.

**Table 1 antioxidants-14-00395-t001:** Survey-weighted baseline characteristics of children and adolescents by anemia status in NHANES 2007–2010 and 2017–2018.

Charasteristics	All Subjects (N = 6815)	Non-Anemia (N = 6505)	Anemia (N = 310)	*p* ^a^
**Age**				<0.001
1–6	2049 (23.69)	1975 (23.84)	74 (19.85)	
7–12	2177 (29.26)	2108 (29.70)	69 (18.26)	
13–20	2589 (47.05)	2422 (46.46)	167 (61.89)	
**Sex**				<0.001
Female	3336 (49.0)	3116 (48)	220 (74.06)	
Male	3479 (51.0)	3389 (52)	90 (25.94)	
Energy intake(kcal)	1788.50 (1409.00, 2258.50)	1791.50 (1414.00, 2263.50)	1694.00 (1314.50, 2161.00)	0.05
Vitamin C(mg)	77.49(74.48, 80.49)	77.55(74.44, 80.65)	76.02(68.47, 83.57)	0.71
Vitamin E(mg)	6.58(6.39, 6.76)	6.59(6.40, 6.77)	6.33(5.83, 6.83)	0.36
Vitamin B12(μg)	4.29 (2.81, 6.19)	4.32 (2.83, 6.24)	3.80 (2.42, 4.91)	<0.001
Vitamin B6(mg)	1.55 (1.13, 2.13)	1.56 (1.14, 2.15)	1.37 (1.01, 1.91)	<0.001
Total folate(μg)	327.50 (237.50, 455.50)	328.00 (238.00, 456.50)	290.00 (204.50, 406.50)	<0.001
Iron(mg)	12.60 (9.28, 17.24)	12.65 (9.31, 17.33)	11.21 (8.36, 15.11)	<0.001
Total sum of all 29 flavonoids(mg)	40.10 (18.41, 87.10)	40.41 (18.53, 87.56)	32.95 (14.52, 82.44)	0.03
BMI				<0.001
Normal weight	5071 (75.08)	4868 (78.71)	203 (71.95)	
Overweight	736 (12.20)	702 (12.80)	34 (11.36)	
Obese	555 (8.42)	509 (8.49)	46 (16.68)	
Ethnicity				<0.001
White	2220 (55.84)	2175 (56.72)	45 (33.73)	
Black	1494 (13.45)	1329 (12.40)	165 (39.67)	
Other	3101 (30.71)	3001 (30.88)	100 (26.51)	
PIR				0.06
≤1.35	2965 (32.69)	2817 (32.33)	148 (41.60)	
1.36–3	1781 (25.90)	1706 (26.05)	75 (22.10)	
>3	1548 (35.20)	1482 (35.34)	66 (31.64)	
Unknown	521 (6.22)	500 (6.28)	21 (4.67)	

^a^: *p* values were determined from Wilcoxon rank-sum test to compare differences between continuous variables and from Pearson test for categorical variables between cases and non-cases.

**Table 2 antioxidants-14-00395-t002:** ORs (95% CIs) of the prevalence of anemia according to dietary flavonoid intake levels (mg/day) among children and adolescents in NHANES 2007–2010 and 2017–2018.

Flavonoid Class	Low-Intake (Ref)	Median-Intake	High-Intake	*p*-Trend
**Isoflavones**				
Crude	1.00	1.164 (0.794,1.706)	0.933 (0.684, 1.273)	0.425
Model 1	1.00	1.188 (0.799, 1.768)	0.936 (0.684, 1.283)	0.412
Model 2	1.00	1.253 (0.803, 1.955)	1.005 (0.701, 1.442)	0.68
Model 3	1.00	1.266 (0.791, 2.026)	1.007 (0.712, 1.423)	0.69
**Anthocyanidins**				
Crude	1.00	0.785 (0.544, 1.134)	0.642 (0.449, 0.917)	**0.035**
Model 1	1.00	0.829 (0.563, 1.223)	0.626 (0.428, 0.917)	**0.026**
Model 2	1.00	0.771 (0.513, 1.158)	0.708 (0.486, 1.031)	0.154
Model 3	1.00	0.724 (0.494, 1.060)	0.992 (0.932, 1.055)	0.179
**Flavan-3-ols**				
Crude	1.00	0.709 (0.478, 1.052)	0.586 (0.430, 0.797)	**0.004**
Model 1	1.00	0.731 (0.492, 1.085)	0.580 (0.424, 0.793)	**0.003**
Model 2	1.00	0.806 (0.514, 1.264)	0.624 (0.414, 0.940)	**0.03**
Model 3	1.00	0.816 (0.517, 1.289)	0.612 (0.406, 0.921)	**0.023**
**Flavanones**				
Crude	1.00	0.713 (0.531, 0.955)	0.835 (0.643, 1.084)	0.724
Model 1	1.00	0.680 (0.511, 0.905)	0.820 (0.632, 1.065)	0.746
Model 2	1.00	0.668 (0.497, 0.898)	0.720 (0.509, 1.017)	0.345
Model 3	1.00	0.664 (0.496, 0.890)	0.747 (0.505, 1.105)	0.493
**Flavones**				
Crude	1.00	0.785 (0.551, 1.117)	0.865 (0.615, 1.217)	0.578
Model 1	1.00	0.759 (0.527, 1.093)	0.807 (0.566, 1.150)	0.364
Model 2	1.00	0.668 (0.497, 0.898)	0.720 (0.509, 1.017)	0.932
Model 3	1.00	0.999 (0.687, 1.455)	0.990 (0.931, 1.053)	0.752
**Flavonols**				
Crude	1.00	1.069 (0.767, 1.491)	0.790 (0.561, 1.114)	**0.01**
Model 1	1.00	1.035 (0.739, 1.449)	0.720 (0.519, 0.999)	0.06
Model 2	1.00	1.056 (0.698, 1.597)	0.790 (0.516, 1.209)	0.29
Model 3	1.00	1.064 (0.697, 1.625)	0.813 (0.522, 1.267)	0.24
**Total flavonoids**				
Crude	1.00	0.682 (0.485, 0.959)	0.684 (0.511, 0.915)	**0.035**
Model 1	1.00	0.688 (0.479, 0.986)	0.642 (0.483, 0.853)	**0.008**
Model 2	1.00	0.662 (0.438, 0.999)	0.657 (0.465, 0.929)	**0.045**
Model 3	1.00	0.655 (0.428, 1.003)	0.641 (0.439, 0.935)	**0.044**

Four models are presented: Crude (unadjusted); Model 1, adjusted for age (1–6, 7–12, 13–20 years) and sex (male, female); Model 2, further adjusted for ethnicity (White, Black, Other), Poverty Income Ratio (PIR: ≤1.35, 1.36–3.0, >3.0), energy intake (tertiles), and BMI (normal weight, overweight, obese); and Model 3, additionally adjusted for the intake of vitamins B12, B6, folate, C, E, and iron (tertiles). Low intake is the reference category (OR = 1.00). *p*-trend values assess the trend across tertiles within each model.

**Table 3 antioxidants-14-00395-t003:** Stratified analyses of the prevalence of anemia according to dietary total flavonoid intake levels (mg/day) among children and adolescents in NHANES 2007–2010 and 2017–2018.

	Low-Intake (Ref)	Median-Intake	High-Intake	*p* for Trend	*p* for Interaction
**Age**					0.919
1–6	1.00	0.905 (0.480, 1.703)	0.712 (0.363, 1.395)	0.307	
7–12	1.00	0.577 (0.303, 1.100)	0.637 (0.332, 1.221)	0.149	
13–20	1.00	0.711 (0.411, 1.230)	0.664 (0.440, 0.997)	**0.048**	
**Sex**					**0.002**
Female	1.00	0.475 (0.297, 0.760)	0.668 (0.475, 0.939)	**0.025**	
Male	1.00	1.426 (0.835, 2.436)	0.616 (0.345, 1.101)	0.113	
**Ethnicity**					0.35
White	1.00	0.573 (0.251, 1.308)	0.897 (0.488, 1.649)	0.701	
Black	1.00	0.736 (0.465, 1.166)	0.721 (0.508, 1.024)	0.069	
Other	1.00	0.636 (0.364, 1.110)	0.436 (0.266, 0.714)	**0.002**	
**PIR**					0.363
>3	1.00	0.739 (0.372, 1.469)	1.027 (0.563, 1.872)	0.915	
1.36–3	1.00	0.574 (0.285, 1.155)	0.374 (0.168, 0.835)	**0.019**	
≤1.35	1.00	0.752 (0.485, 1.166)	0.713 (0.446, 1.142)	0.158	
Unknown	1.00	0.417 (0.106, 1.650)	0.418 (0.120, 1.464)	0.171	
**BMI**					0.122
Normal weight	1.00	0.612 (0.394, 0.951)	0.745 (0.524, 1.060)	0.098	
Obese	1.00	0.606 (0.258, 1.423)	0.366 (0.161, 0.835)	**0.016**	
Overweight	1.00	1.709 (0.564, 5.178)	0.709 (0.234, 2.144)	0.57	

Odds ratios (ORs) with 95% confidence intervals (CIs) for anemia prevalence across tertiles of dietary total flavonoid intake (mg/day), stratified by age (1–6, 7–12, 13–20 years), sex (male, female), ethnicity (White, Black, Other), Poverty Income Ratio (PIR: ≤1.35, 1.36–3.0, >3.0, unknown), and BMI (normal weight, overweight, obese). Low intake is the reference category (OR = 1.00). Analyses are adjusted for age, sex, ethnicity, PIR, energy intake (tertiles), BMI, and intake of vitamins B12, B6, folate, and iron (tertiles).

**Table 4 antioxidants-14-00395-t004:** Stratified analyses of the prevalence of anemia according to dietary flavan-3-ol intake levels (mg/day) among children and adolescents in NHANES 2007–2010 and 2017–2018.

	Low-Intake (Ref)	Median-Intake	High-Intake	*p* for Trend	*p* for Interaction
Age					0.454
1–6	1.00	1.055 (0.506, 2.199)	0.771 (0.440, 1.352)	0.211	
7–12	1.00	0.500 (0.237, 1.055)	0.578 (0.293, 1.141)	0.255	
13–20	1.00	0.842 (0.530, 1.339)	0.563 (0.346, 0.916)	**0.023**	
Sex					0.5
Female	1.00	0.613 (0.396, 0.948)	0.555 (0.390, 0.789)	**0.01**	
Male	1.00	0.932 (0.458, 1.899)	0.650 (0.322, 1.312)	0.181	
Ethnicity					0.367
White	1.00	0.644 (0.233, 1.779)	0.591 (0.258, 1.353)	0.292	
Black	1.00	1.147 (0.767, 1.716)	0.861 (0.616, 1.204)	0.219	
Other	1.00	0.582 (0.358, 0.944)	0.397 (0.224, 0.700)	**0.008**	
PIR					0.497
>3	1.00	0.894 (0.382, 2.092)	0.714 (0.331, 1.541)	0.366	
1.36–3	1.00	0.471 (0.238, 0.933)	0.347 (0.166, 0.727)	**0.018**	
≤1.35	1.00	0.822 (0.520, 1.299)	0.696 (0.432, 1.123)	0.183	
Unknown	1.00	0.302 (0.080, 1.148)	0.533 (0.171, 1.657)	0.458	
BMI					0.351
Normal weight	1.00	0.653 (0.400, 1.067)	0.544 (0.356, 0.831)	**0.019**	
Obese	1.00	0.733 (0.376, 1.427)	0.478 (0.190, 1.201)	0.158	
Overweight	1.00	1.840 (0.723, 4.681)	1.163 (0.402, 3.368)	0.933	

Odds ratios (ORs) with 95% confidence intervals (CIs) for anemia prevalence across tertiles of dietary flavan-3-ol intake (mg/day), stratified by age (1–6, 7–12, 13–20 years), sex (male, female), ethnicity (White, Black, Other), Poverty Income Ratio (PIR: ≤1.35, 1.36–3.0, >3.0, unknown), and BMI (normal weight, overweight, obese). Low intake is the reference category (OR = 1.00). Analyses are adjusted for age, sex, ethnicity, PIR, energy intake (tertiles), BMI, and intake of vitamins B12, B6, folate, and iron (tertiles).

## Data Availability

The data used in this manuscript are publicly available at the NHANES website: https://wwwn.cdc.gov/nchs/nhanes/ (accessed on 5 January 2025); and the FNDD website: https://www.ars.usda.gov/northeast-area/beltsville-md-bhnrc/beltsville-human-nutrition-research-center/food-surveys-research-group/docs/fndds-flavonoid-database/ (accessed on 5 January 2025).

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
