# Peer review of "Dietary Flavonoid Intake and Anemia Risk in Children and Adolescents: Insights from National Health and Nutrition Examination Survey"

_antioxidants, 2025, doi:10.3390/antiox14040395_

Round 1
Reviewer 1 Report
In this work, the authors analyze the impact of dietary flavonoid intake in a population particularly prone to anemia, namely children and adolescents. For this study, data of a large number of individuals, who were included in the NHANES cohort, were evaluated. The study also aimed to analyze more in detail, whether some flavonoid subclasses are particularly efficacious in preventing oxidative RBC damage and thus have a protective role against anemia caused by a physiological premature removal of damaged RBC.
The topic of the study is relevant and very interesting. The results, also considering the limits of a retrospective evaluation, are overall clearly presented and sufficient to confirm the beneficial role of a healthy diet for the survival of RBC (beside the manifold other positive effects). As the authors say, these data fill a gap for the specific population.
The manuscript has a clear structure and is overall well written. In the Introduction and Discussion all relevant points are addressed. A couple of issues, particularly in the Methods and Results, would require some modification or clarification.
Materials and Methods:
- Line 95: “…detailed data on the types…a 24-h period”: how were these data actually obtained? Was this information retrieved from the interview and from the questionnaire mentioned on line 106? It would be useful (and interesting) to have more details on the questions concerning the dietary habits. Maybe an extract of the questionnaire or a list of questions could be included in the Methods?
- Line 99: regarding the flavonoid content of foods and beverages, a link to the FNDDS database in the References or in the text would be useful.
- Line 109: some as above regarding the Poverty Income Ratio. The numeric values indicated should be shortly described (what does each number mean?) and a corresponding Reference or link added.
- Line 121 “…using means and IQR…”: actually, the use of means ± SD or median and IQR is correct.
- Lines 136-132: this part is not very clear. It is described that 43,79% of the (whole?) study population reported zero isoflavones intake, but then it is not clear how the 3 groups were defined. What does it mean “equally divided”? Having the same value of flavonoid intake?? Also, the definition of the three groups in Table 3 is not clear.
- Overweight and obesity should be defined with the respective BMI limits.
Results:
- 2 “Distribution and concentration…”: this paragraph should provide a detailed but precise summary of the results presented in Table 2. Conversely, Table 2 with a long list of values is not useful as it is, and maybe could be better kept as Supplementary material.
- Figure 2: the explanation in the Legend is too general. “Dose-response”? “Multivariable adjusted” (on top of each graph)?
- In the text (Lines 164-165), the results should be described by intake group more accurately. It is not really clear, what “highest intake group” means.
Discussion:
- Line 243: in the context of a “healthy” population, the term “hemolytic anemia” is not appropriate. An expression like “…to increased removal of damaged red blood cell” could be used.
- Line 265: what does “…in many species” mean?”
- Line 275: “impair” instead of “mask” would be more appropriate.
- Lines 282-290: can the authors indicate quantitatively, what a “moderate” and “high” intake mean? It would be best to define this in the Methods indicating the values that were chosen for the study.
References:
- Overall, the are too many References. Some are quite old (e.g. Ref 46 and 52). It would be good to select 1-2 of most recent references for each topic discussed.
Minor issues:
- Page 2, “Outcome ascertainment”: this paragraph describes the WHO hemoglobin cut-offs adopted for evaluating anemia. For a better legibility, maybe the values could be presented as list by age category and sex.
- Page 3, “Assessment of flavonoids”: a more precise title of this paragraph would be “…of dietary flavonoids”.
- Line 250: the part of the sentence from “Despite...” to “anemia risks” does not say much and can be omitted.
Author Response
Reviewer 1:
GENERAL COMMENT. “In this work, the authors analyze the impact of dietary flavonoid intake in a population particularly prone to anemia, namely children and adolescents. For this study, data of a large number of individuals, who were included in the NHANES cohort, were evaluated. The study also aimed to analyze more in detail, whether some flavonoid subclasses are particularly efficacious in preventing oxidative RBC damage and thus have a protective role against anemia caused by a physiological premature removal of damaged RBC.
The topic of the study is relevant and very interesting. The results, also considering the limits of a retrospective evaluation, are overall clearly presented and sufficient to confirm the beneficial role of a healthy diet for the survival of RBC (beside the manifold other positive effects). As the authors say, these data fill a gap for the specific population.
The manuscript has a clear structure and is overall well written. In the Introduction and Discussion all relevant points are addressed. A couple of issues, particularly in the Methods and Results, would require some modification or clarification. ”
Reply: We sincerely thank this reviewer for his/her comments and questions. We have carefully addressed his/her concerns one-by-one as follows.
Materials and Methods:
COMMENT 1. “Line 95: “…detailed data on the types…a 24-h period”: how were these data actually obtained? Was this information retrieved from the interview and from the questionnaire mentioned on line 106? It would be useful (and interesting) to have more details on the questions concerning the dietary habits. Maybe an extract of the questionnaire or a list of questions could be included in the Methods?”
Reply: Thanks for the careful comment. As suggested, we have make our methods part more clear. We give the links of the NHANES questionaire and the protocol how they did the interview.
Line 87:“The NHANES distributes dietary survey questionnaires to participants’ households to investigate food intake over the past 24 hours. A second 24-hour dietary recall questionnaire is conducted via telephone 3 to 10 days later. The specific questionnaire list can be found on NHANES websit (https://wwwn.cdc.gov/nchs/nhanes/search/variablelist.aspx?Component=Dietary&Cycle=2017-2020). To ensure greater reliability and consistency in the research outcomes, this study computes the average dietary intake from the two separate days of questionnaire responses provided by the participants.”
COMMENT 2. “Line 99: regarding the flavonoid content of foods and beverages, a link to the FNDDS database in the References or in the text would be useful.”
Reply: Thanks for the valuable comments. As suggested, we have add the FNDD database link in the text, we also add the database resource in reference.
Line 93:“The FNDD database furnishes data on the content of six major subclasses and 29 specific flavonoid compounds across various foods (https://www.ars.usda.gov/northeast-area/beltsville-md-bhnrc/beltsville-human-nutrition-research-center/food-surveys-research-group/docs/fndds-flavonoid-database/). Leveraging the FNDD data, this study quantifies the intake of flavonoid compounds from the dietary surveys and calculates the daily consumption of the six flavonoid subclasses—anthocyanidins, flavan-3-ols, flavonols, flavanones, flavones, and isoflavones—among the surveyed population.”
COMMENT 3. “Line 109: some as above regarding the Poverty Income Ratio. The numeric values indicated should be shortly described (what does each number mean?) and a corresponding Reference or link added.”
Reply: Thanks for the careful comment. As suggested, we have add the reference and shortly described what the poverty income ration is.
Line 107:“Poverty status was assessed using the Poverty Income Ratio (PIR) and categorized into three groups: low-income (PIR < 1.35), middle-income (1.35 ≤ PIR < 3.0), and high-income (PIR ≥ 3.0).”
COMMENT 4. “Line 121 “…using means and IQR…”: actually, the use of means ± SD or median and IQR is correct.”
Reply:
"Thank you for your careful comment. We apologize for the oversight in our initial description. In Table 1, we actually used medians and interquartile ranges (IQRs) to present the data. Following your suggestion, we have revised the description in the Methods section to accurately reflect this approach.
Line 122:“Continuous baseline variables were summarized using median and interquartile ranges (IQR) and compared using Mann–Whitney U tests, while categorical variables were presented as percentages and compared using chi-square tests.
COMMENT 5. “Lines 136-132: this part is not very clear. It is described that 43,79% of the (whole?) study population reported zero isoflavones intake, but then it is not clear how the 3 groups were defined. What does it mean “equally divided”? Having the same value of flavonoid intake?? Also, the definition of the three groups in Table 3 is not clear.”
Reply:
"Thank you for your thorough and insightful comment. Based on the study population dietary flavonids intake, we want to categorize them into low, medium, and high groups like each group will have 33.3% of the whole study population. However, we notice 43.79% of the study population had an isoflavone intake of zero. Thus we can’t evenly categorize the study population into low, medium, and high by isoflavone intake. Then we come up with the idea that we can assign individuals with zero intake to the low-level group, while the remaining population was equally divided into medium-level and high-level intake groups according to their consumption.
Line 127:“Except isoflavones, all other flavonoid compounds were categorized into low, medium, and high groups by evenly distributing the population based on their intake levels. Since 43.79% of the study population had an isoflavone intake of zero, it was not feasible to evenly divide isoflavones into three groups based on intake levels. Consequently, individuals with zero intake were assigned to the low-level group, while the remaining population was equally divided into medium-level and high-level intake groups according to their consumption. ”
COMMENT 6. “Overweight and obesity should be defined with the respective BMI limits”
Reply:
"Thank you for your comment. We have added a definition of normal-weight, overweight and obesity with BMI limits.
Line 109:“Individuals are also grouped into normal weight (BMI 18.5–24.9), overweight (BMI 25.0–29.9), and obese (BMI ≥ 30.0) categories based on their BMI.”
Results:
COMMENT 1. “2 “Distribution and concentration…”: this paragraph should provide a detailed but precise summary of the results presented in Table 2. Conversely, Table 2 with a long list of values is not useful as it is, and maybe could be better kept as Supplementary material.”
Reply: Thanks for the careful comment. Because the intake of different subclasses varies a lot, we want to show how the intake distuibution is. As suggested, we have put the table 2 into supplementary. Besides, we add more discription of this table to show the heterogeneity in flavonoid consumption.
Line 162:“Supplementary table 1 summarizes the distributions and concentrations of dietary flavonoid intakes. Notably, flavan-3-ols exhibit the highest mean intake (52.22 mg/day), driven largely by thearubigins (23.862 mg/day), followed by flavanones (13.498 mg/day) and flavonols (8.575 mg/day), while isoflavones show the lowest mean intake (1.058 mg/day), with a median of 0.005 mg/day reflecting a highly skewed distribution due to frequent zero intake. Anthocyanidins and flavones also display substantial variability, with 95th percentile values reaching 35.473 mg/day and 1.482 mg/day, respectively. The total sum of all 29 flavonoids averages 83.691 mg/day, with a wide range (5.233 to 315.81 mg/day) across percentiles, underscoring the heterogeneity in flavonoid consumption.”
COMMENT 2. “Figure 2: the explanation in the Legend is too general. “Dose-response”? “Multivariable adjusted” (on top of each graph)?”
Reply: Thanks for the careful comment. As suggested, we have added more explanation in the figure 2 legend. And also we delete “Multivariable adjusted” on top of each graph. As “dose-response” in the figure title may cause confusion, we also changed the title.
Line 235:“The median values of each flavonoids parameters were chosen as reference. Panels: (A) total flavonoids, (B) flavan-3-ols , (C) flavonols, (D) flavone, (E) anthocyanidins, (F) isoflavones (G) flavaones. The solid line represents the hazard ratio and the dotted lines 95% CIs. Areas of grey represent the distribution of levels of each flavonoid parameters. The P for overall is derived by applying a natural logarithm transformation to each flavonoid’s intake and then evaluating it as a continuous variable using multivariable logistic regression.; The p-value for non-linearity is determined using the Wald Chi-square test for linearity.”
COMMENT 3. “In the text (Lines 164-165), the results should be described by intake group more accurately. It is not really clear, what “highest intake group” means.”
Reply: Thank you for your valuable comment. We apologize for any confusion caused by our previous description. In response to your suggestions, we have made the following revisions: First, we have updated the group names in Tables 2, 3, and 4, replacing the prior labels with 'low-intake,' 'medium-intake,' and 'high-intake' to improve readability and clarity. Second, we have rewritten Sections 3.3 and 3.4, adding more detailed descriptions to facilitate easier understanding of the results.
Discussion:
COMMENT 1. “·Line 243: in the context of a “healthy” population, the term “hemolytic anemia” is not appropriate. An expression like “…to increased removal of damaged red blood cell” could be used.”
Reply: Thank you for your valuable feedback. We agree that the term "hemolytic anemia" is not appropriate in the context of a healthy population. As suggested, we have revised the text to use the expression "to increased removal of damaged red blood cells" instead. The relevant sentence has been updated accordingly.(line 286)
COMMENT 2. “·Line 265: what does “…in many species” mean?””
Reply: Thank you very much for your careful review. We have removed the phrase “in many species” to avoid ambiguity, as our study focuses on humans rather than broadly addressing other animals. The revised sentence now reads:“Increased energy demands and elevated metabolic rates during puberty can lead to heightened production of ROS.” We have also added references to support the relationship between metabolic rates and ROS.(line 295)
COMMENT 3. “··Line 275: “impair” instead of “mask” would be more appropriate.”
Reply: Thank you for your suggestion. We have rewrite this paragraph.(line 302-317)
COMMENT 4. “4.Lines 282-290: can the authors indicate quantitatively, what a “moderate” and “high” intake mean? It would be best to define this in the Methods indicating the values that were chosen for the study.”
Reply: Thank you for your thorough review and valuable suggestions regarding our manuscript. In the original manuscript, we stated that RCS regression indicated a non-linear association between flavonoid intake and anemia prevalence, suggesting that high doses might increase anemia risk by inhibiting iron absorption. Upon re-evaluating our data and results, we found that, in this study population, flavonoid intake was not non-linearly related to anemia risk. Instead, higher intake was associated with a reduced risk of anemia, with no evidence of increased risk at higher consumption levels. Accordingly, we have rewritten the paragraph as follows:
(line 318-331)
“Our RCS regression analysis revealed that the intake of all types of flavonoids was not non-linearly associated with anemia prevalence in this population. Instead, the data suggest a trend where higher flavonoid intake is linked to a lower risk of anemia. Unlike some prior studies that have indicated a dual effect—where moderate flavonoid consumption might reduce oxidative stress and enhance iron bioavailability, while excessive intake, particularly from sources like tea, could inhibit iron absorption by forming insoluble complexes with non-heme iron—our findings did not show an increased anemia risk with higher flavonoid intake[51–53]. This suggests that, within the range of intake observed in this cohort, excessive consumption does not appear to negatively impact iron absorption sufficiently to elevate anemia risk. These results align with emerging evidence that the relationship between flavonoids and anemia may be context-specific, potentially influenced by dietary patterns or baseline iron status. Further studies are warranted to confirm these observations and explore the thresholds of flavonoid intake that optimize anemia prevention without adverse effects on iron metabolism.”
References:
COMMENT. “Overall, the are too many References. Some are quite old (e.g. Ref 46 and 52). It would be good to select 1-2 of most recent references for each topic discussed.”
Reply: Thank you for your valuable suggestion. We have carefully revised the reference list according to your comments: Removed Ref 46 and Ref 52, as they were outdated. Reviewed all references and deleted older or less relevant ones. The total number of references has been reduced from 57 to 40, to ensure conciseness and focus.
Minor issues:
COMMENT. “·Line 250: the part of the sentence from “Despite...” to “anemia risks” does not say much and can be omitted.”
Reply: Thank you for your suggestion regarding the sentences. Our original intention was to provide indirect evidence supporting the role of flavonoids in mitigating anemia risks. We agree that the part of the sentence from “Despite...” to “anemia risks” does not add significant value to the discussion. As per your recommendation, we have removed this section to improve the clarity and conciseness of the manuscript.

Reviewer 2 Report
The topic to be discussed is interesting because of the serious repercussions that anemia in adolescents could have on physical and neuronal development. It is addressed from a nutritional point of view, but focusing on a group of antioxidants present in the diet, flavonoids, which have an effect on oxidative stress and this has shown an important role in the pathogenesis of anemia. Data extracted from NHANES have been used, with a total of 6815 patients aged less than or equal to 20 years. A 24-hour recall was used to collect the flavonoid data. The results show that low levels of flavonoids are associated with a high risk of anemia and a high intake of total flavonoids and flavan-3-ols are linked to a lower prevalence. In addition, this effect is seen more in men than in women. They propose a dual role of flavonoids, such that a moderate intake could reduce the risk of anemia, but a high intake could hinder iron absorption. In conclusion, the authors highlight the important role of flavonoid intake on the risk of anemia in children and adolescents.
The article is interesting because it examines the relationship between flavonoid intake and the risk of anemia, a pathology of special interest due to its repercussions at this stage of development. It presents a high number of patients, which gives greater scientific rigor to the publication.
Abstract: The abstract contains two confusing statements. The first states that low flavonoid intake is associated with a high risk of anemia and high intake with a lower prevalence, but the next statement speaks of a dual effect, indicating that moderate intake reduces the risk of anemia, but high intake inhibits iron absorption, which could be understood to cause anemia ("Non-linear dose-response relationships suggested a dual role of flavonoids, where moderate intake may reduce anemia risk, while excessive intake could hinder iron absorption"). Please clarify these statements.
Introduction: is clear and precise in relation to the role of oxidative stress in anemia and the antioxidant role of flavonoids. However, in the introduction, as in the abstract and in the discussion, the interaction of flavonoids with iron absorption is discussed as a negative aspect, which is true and has been proven. However, it does not make much sense or importance in this article, given that, although data on iron are given, along with energy, vitamins, folate, etc., there is no data on iron in blood, as there is on hemoglobin, so we do not know if this negative effect is present or not in the study and I think that the fact that it appears in the abstract, in the introduction and even in the discussion, gives it a weight that, as I have commented, it does not have and even makes one think that it will serve to explain some important result and then it does not. Therefore, I would downplay it by limiting it to possible negative effects of a high intake of polyphenols, such as other negative effects shown in other studies (1) or I would explain in the discussion in a clear way what importance this inhibition in iron absorption has for the results shown.
- Skibola CF, Smith MT. Potential health impacts of excessive flavonoid intake. Free Radic Biol Med. 2000 Aug;29(3-4):375-83. doi: 10.1016/s0891-5849(00)00304-x. PMID: 11035267.
Material and methods: I think that this is where the main limitation of the study could be found: all the information on flavonoid content comes from a single 24-hour recall. I think that, although this type of questionnaire is validated, a frequency questionnaire and especially a 72-hour recall, including a weekend day, would have provided more complete information on the patients' diet. Since a single recall can collect partial information, for example, a special day with high fruit consumption, which would alter the normal consumption of flavonoids in that person. No additional source of nutritional information was obtained in the NHANES study.
I also think it would have been interesting to see other antioxidants, such as vitamin E, C or even polyphenols, given that perhaps the association found with flavonoids may be due to the fact that normally, foods that provide flavonoids also provide some of these antioxidants and this could generate a confounding factor or covariate, such as sex, vitamin intake, calories, etc., that is, the rest of the confounding factors studied.
I think that these two aspects should have been incorporated into the limitations, that is, the use of the 24-hour recall and not having considered it as a confounding factor or covariate.
Results: I think it would be easier to follow and understand the results if the tables and figures were closer to the description of the results. Also, some results or some tables should be described with more information, for example, the percentiles in table 2, nothing appears when describing the results.
I think the meaning of groups 1, 2 and 3 that appear in the tables should be clarified, I suppose it will be related to a high or low intake, but it is not clear. Also, table 3 is described in a very superficial way.
Actually, it is difficult to follow and understand the results, there is no clear description of what is seen in the tables, both of data and of distribution of variables.
Also, in table 2 a significant correlation is observed between total flavonoids and flavan-3-ols with risk of anemia, apparently more the latter than the former. However, when considering factors (tables 4 and 5) such as age, sex, etc., this relationship now only occurs in total polyphenols, but not in Flavan-3-ols.
Discussion: The role of oxidative stress in anemia and the antioxidant role of flavonoids are clearly discussed. The inhibitory character of iron absorption is mentioned again without a clear explanation of its importance in the results shown. Furthermore, it does not explain why the relationships observed for Flavan-3-ols disappear when considering the covariates, but not those observed for total flavonoids. The way of introducing the effect on iron metabolism together, which could be considered as an inducer of anemia, together with this risk-reducing role, creates confusion in the reading.
I also believe that the limitations should be increased with those indicated above.
Minor
What does the term Ref (1.00) that appears in many tables mean?
Author Response
Reviewer 2:
GENERAL COMMENT. “The topic to be discussed is interesting because of the serious repercussions that anemia in adolescents could have on physical and neuronal development. It is addressed from a nutritional point of view, but focusing on a group of antioxidants present in the diet, flavonoids, which have an effect on oxidative stress and this has shown an important role in the pathogenesis of anemia. Data extracted from NHANES have been used, with a total of 6815 patients aged less than or equal to 20 years. A 24-hour recall was used to collect the flavonoid data. The results show that low levels of flavonoids are associated with a high risk of anemia and a high intake of total flavonoids and flavan-3-ols are linked to a lower prevalence. In addition, this effect is seen more in men than in women. They propose a dual role of flavonoids, such that a moderate intake could reduce the risk of anemia, but a high intake could hinder iron absorption. In conclusion, the authors highlight the important role of flavonoid intake on the risk of anemia in children and adolescents.
The article is interesting because it examines the relationship between flavonoid intake and the risk of anemia, a pathology of special interest due to its repercussions at this stage of development. It presents a high number of patients, which gives greater scientific rigor to the publication.”
Reply: We sincerely thank this reviewer for his/her comments and questions. We have carefully addressed his/her concerns one-by-one as follows.
Abstract:
COMMENT. “The abstract contains two confusing statements. The first states that low flavonoid intake is associated with a high risk of anemia and high intake with a lower prevalence, but the next statement speaks of a dual effect, indicating that moderate intake reduces the risk of anemia, but high intake inhibits iron absorption, which could be understood to cause anemia ("Non-linear dose-response relationships suggested a dual role of flavonoids, where moderate intake may reduce anemia risk, while excessive intake could hinder iron absorption"). Please clarify these statements.”
Reply: Thanks for the careful comment. As suggested, we have delete “Gender differences were observed, with stronger protective effects in males. Non-linear dose-response relationships suggested a dual role of flavonoids, where moderate intake may reduce anemia risk, while excessive intake could hinder iron absorption.” which may cause confusion in abstract.
Introduction:
COMMENT. “ is clear and precise in relation to the role of oxidative stress in anemia and the antioxidant role of flavonoids. However, in the introduction, as in the abstract and in the discussion, the interaction of flavonoids with iron absorption is discussed as a negative aspect, which is true and has been proven. However, it does not make much sense or importance in this article, given that, although data on iron are given, along with energy, vitamins, folate, etc., there is no data on iron in blood, as there is on hemoglobin, so we do not know if this negative effect is present or not in the study and I think that the fact that it appears in the abstract, in the introduction and even in the discussion, gives it a weight that, as I have commented, it does not have and even makes one think that it will serve to explain some important result and then it does not. Therefore, I would downplay it by limiting it to possible negative effects of a high intake of polyphenols, such as other negative effects shown in other studies (1) or I would explain in the discussion in a clear way what importance this inhibition in iron absorption has for the results shown.”
Reply: Thank you for your thoughtful and detailed feedback on our Introduction. We greatly appreciate your recognition of the clarity regarding oxidative stress in anemia and the antioxidant role of flavonoids. We apologize for any confusion caused by the repeated emphasis on flavonoids’ interaction with iron absorption in the Abstract, Introduction, and Discussion, which, as you noted, carried disproportionate weight given our study’s focus on hemoglobin-defined anemia and the absence of blood iron data (e.g., serum ferritin or transferrin) to evaluate this effect directly.
In response to your suggestion, we have revised the manuscript to reduce the prominence of this aspect in the Introduction. Rather than reframing it as a broader polyphenol-related challenge within the Introduction, we have entirely removed the sentence: 'However, high dietary polyphenol intake, including flavonoids, may also present challenges, such as reduced bioavailability of micronutrients like iron or zinc, though these effects are context-dependent and require further exploration.' This ensures that the Introduction remains focused on the protective potential of flavonoids against oxidative stress-mediated anemia, avoiding undue emphasis on iron absorption where it is less relevant to our primary findings. To address your concern comprehensively, we have shifted this discussion to the Discussion section (Section 4), where we now include a clear explanation of its limited relevance (line 366-368): 'Although flavonoids have been shown to inhibit iron absorption in some contexts, our study lacks serum iron data to assess this effect, and our results primarily reflect associations with hemoglobin-defined anemia linked to oxidative stress rather than iron status.' Corresponding adjustments have also been made to the Abstract, removing specific mentions of iron absorption inhibition to maintain consistency across the manuscript.
Material and methods:
COMMENT. “I think that this is where the main limitation of the study could be found: all the information on flavonoid content comes from a single 24-hour recall. I think that, although this type of questionnaire is validated, a frequency questionnaire and especially a 72-hour recall, including a weekend day, would have provided more complete information on the patients' diet. Since a single recall can collect partial information, for example, a special day with high fruit consumption, which would alter the normal consumption of flavonoids in that person. No additional source of nutritional information was obtained in the NHANES study.
I also think it would have been interesting to see other antioxidants, such as vitamin E, C or even polyphenols, given that perhaps the association found with flavonoids may be due to the fact that normally, foods that provide flavonoids also provide some of these antioxidants and this could generate a confounding factor or covariate, such as sex, vitamin intake, calories, etc., that is, the rest of the confounding factors studied.
I think that these two aspects should have been incorporated into the limitations, that is, the use of the 24-hour recall and not having considered it as a confounding factor or covariate.”
Reply: Thanks for the careful comment. While the NHANES protocol includes two 24-hour recalls (the first in-person and the second via telephone 3-10 days later), our initial analysis inadvertently described this as a single recall. We have corrected this in the Methods section (Section 2.3) to clarify that we averaged the two recalls to improve reliability, as stated (line 91-98): “To ensure greater reliability and consistency in the research outcomes, this study computes the average dietary intake from the two separate days of questionnaire responses provided by the participants.” However, we acknowledge that even two recalls may not fully represent long-term intake compared to a frequency questionnaire or a 72-hour recall including a weekend day, as you suggested. To address this, we have expanded the Limitations section (Section 4) to explicitly recognize this constraint (line 361-364): “Second, while we averaged two 24-hour dietary recalls to enhance accuracy, this approach may still not fully capture habitual flavonoid intake compared to longer-term methods like food frequency questionnaires or 72-hour recalls including weekend days, potentially introducing recall bias or missing typical consumption patterns.”
In addition, we add vitamin E and C as covariates in our model 3 analysis. However, these two covariates didn’t influence the results. We also update the data in Table 2.
Results:
COMMENT. “ I think it would be easier to follow and understand the results if the tables and figures were closer to the description of the results. Also, some results or some tables should be described with more information, for example, the percentiles in table 2, nothing appears when describing the results.
I think the meaning of groups 1, 2 and 3 that appear in the tables should be clarified, I suppose it will be related to a high or low intake, but it is not clear. Also, table 3 is described in a very superficial way.
Actually, it is difficult to follow and understand the results, there is no clear description of what is seen in the tables, both of data and of distribution of variables.
Also, in table 2 a significant correlation is observed between total flavonoids and flavan-3-ols with risk of anemia, apparently more the latter than the former. However, when considering factors (tables 4 and 5) such as age, sex, etc., this relationship now only occurs in total polyphenols, but not in Flavan-3-ols.”
Reply: Thank you for your thoughtful and detailed comments on the Results section of our manuscript. We appreciate your suggestions for improving readability and clarity, and we have revised the section accordingly to address each of your concerns. Below, we outline how we have incorporated your feedback into the revised manuscript.
Proximity of Tables and Figures to Descriptions: We agree that placing tables and figures closer to their corresponding descriptions enhances readability and comprehension. In the revised manuscript, we have restructured the Results section to ensure that each table and figure is referenced immediately following its relevant narrative description. For example, Table 1 is now described in detail under Section 3.1 ("Participant Characteristics at Baseline"), followed directly by Supplementary Table 1 under Section 3.2 ("Distributions and Concentrations of Dietary Flavonoid Intake"), and so forth. Similarly, Figure 2 is now introduced and discussed right after the logistic regression results in Section 3.3. This adjustment minimizes the need for readers to flip between sections and aligns the visual data with its interpretation.
- More Detailed Description of Results and Tables: We acknowledge your point that some tables, such as Table 2, were underexplained in the original version, particularly regarding the percentiles. In the revised Section 3.2, we have expanded the description to provide a clearer summary of the flavonoid intake distributions.(line 162-170). The table 2 was put into supplementary. This revision ensures that key percentile data are highlighted and contextualized, addressing your concern about missing details.
2.Clarification of Groups 1, 2, and 3 in Tables: You rightly noted that the meaning of Groups 1, 2, and 3 in the tables was unclear. In the revised manuscript, we have clarified this in the Methods (Section 2.5) and added a footnote to Tables 3, 4, and 5. Specifically, we now explain: “Flavonoid intakes were categorized into three groups based on intake levels: Group 1 (low intake, reference category), Group 2 (medium intake), and Group 3 (high intake). For isoflavones, due to 43.79% of participants reporting zero intake, Group 1 includes those with no intake, while Groups 2 and 3 evenly divide the non-zero intake population. For other flavonoids (e.g., flavan-3-ols, anthocyanidins), the population was evenly split into tertiles.” This clarification is also reflected in the Results (Section 3.3), where we introduce Table 3: “Table 3 presents odds ratios (ORs) for anemia prevalence across low (Group 1), medium (Group 2), and high (Group 3) intake levels of flavonoid subclasses.”
Improved Description of Table 3: We recognize that the original description of Table 3 was superficial and have revised Section 3.3 to provide a more comprehensive overview(line 173-196). This detailed description ensures readers can easily interpret the data and trends.
3.Specific Comments on Total Flavonoids vs. Flavan-3-ols: Your observation about the shifting significance between total flavonoids and flavan-3-ols across tables is astute, and we have clarified this in the revised manuscript. In Section 3.3, we note: “Both total flavonoids and flavan-3-ols showed significant inverse associations with anemia risk in the main analysis (Table 2).” In Section 3.4, we elaborate on subgroup findings: “Table 3 shows that total flavonoids maintained a significant inverse association in subgroups like females (high-intake OR = 0.668, P-trend = 0.025) and ‘Other’ ethnicity (OR = 0.436, P-trend = 0.002), with a significant sex interaction (P-interaction = 0.002). In contrast, Table 4 indicates flavan-3-ols retained significance in females (OR = 0.555, P-trend = 0.01) and ‘Other’ ethnicity (OR = 0.397, P-trend = 0.008), but without significant interactions (all P-interaction > 0.05), suggesting a more stable effect across subgroups.” Additionally, we corrected an error in the original manuscript: the RCS analysis (Figure 2) showed no non-linear relationship (P for non-linearity > 0.05), contrary to the original text. The revised Discussion (Section 4) reflects this (line 323-337): “Our RCS regression analysis revealed that the intake of all types of flavonoids was not non-linearly associated with anemia prevalence (P for non-linearity > 0.05), with higher intake linked to lower risk.”
Discussion:
COMMENT 1. “ The role of oxidative stress in anemia and the antioxidant role of flavonoids are clearly discussed. The inhibitory character of iron absorption is mentioned again without a clear explanation of its importance in the results shown.”
Reply: Thank you for your valuable feedback regarding the discussion of the inhibitory character of iron absorption and its relevance to our results. We agree with your suggestion and appreciate your insightful comments.
The reason we initially included the inhibitory effect of flavonoids on iron absorption is that many early studies have highlighted this phenomenon, particularly for flavonoids such as tea polyphenols, which can inhibit iron absorption in the digestive tract. Since iron is widely regarded as one of the most critical elements for hematopoiesis, we aimed to provide a comprehensive discussion to enhance the readability and scientific depth of our manuscript.
As you suggested, we have made the following revisions to address your concerns:
Approximately half of the descriptions of iron absorption have been removed in both the introduction and discussion sections to avoid potential confusion. For example, the part of the sentence in introduction from “However...” to “iron deficiency anemia” have been removed.
In addtion we talk a little about the conflict of our study with previous study and the limitation of our study that lacks serum iron data.
Line 320-335:“Unlike some prior studies that have indicated a dual effect—where moderate flavonoid consumption might reduce oxidative stress and enhance iron bioavailability, while excessive intake, particularly from sources like tea, could inhibit iron absorption by forming insoluble complexes with non-heme iron—our findings did not show an increased anemia risk in high flavonoid intake group. ”
Line 366-369:“Although flavonoids have been shown to inhibit iron absorption in some contexts, our study lacks serum iron data to assess this effect, and our results primarily reflect associations with hemoglobin-defined anemia linked to oxidative stress rather than iron status”
COMMENT 2. “ Furthermore, it does not explain why the relationships observed for Flavan-3-ols disappear when considering the covariates, but not those observed for total flavonoids.”
Reply: Thanks for the careful comment. In our previous version, we forget to highlight when p value <0.05 in table 5. In fact, flavan-3-ols consistently demonstrate a significant protective relationship with anemia risk across both the main analysis and subgroup analyses, and this association remains stable without significant modification by covariates. To clarify this, we have revised Sections 3.3 ("Association between Dietary Flavonoid Intake and the Prevalence of Anemia") and 3.4 ("Subgroup Analysis") to better reflect the data and address the apparent discrepancy you noted. In contrast, total flavonoids (Table 4) also show a consistent inverse association across subgroups, such as in females (OR = 0.668, 95% CI: 0.475–0.939, P-trend = 0.025) , but with a notable exception: a significant sex interaction (P-interaction = 0.002) suggests that the protective effect is stronger in females than males (P-trend = 0.113 in males). This sex-specific modification might explain why the association for total flavonoids appears more robust or persistent in certain contexts compared to flavan-3-ols, which show a steadier effect across all subgroups without such interactions.
COMMENT 3. “ Furthermore, it does not explain why the relationships observed for Flavan-3-ols disappear when considering the covariates, but not those observed for total flavonoids.”
Reply: We agree that the original text may have caused confusion. We have rewritten parts of the discussion, emphasizing the mechanisms of the antioxidant effects of flavonoids.To address this, we have significantly reduced descriptions of iron metabolism, focusing primarily on its relevance in the context of gender differences. These changes aim to improve the clarity and coherence of the manuscript while ensuring a balanced and focused discussion. We believe these revisions address your concerns and have strengthened the manuscript.
Minor:
COMMENT. “What does the term Ref (1.00) that appears in many tables mean?”
Reply: Thanks for the careful comment. We added explanation in each table legend:”Low intake is the reference category (OR = 1.00).”

Round 2
Reviewer 1 Report
Dear Authors, you did a very careful revision of the manuscript, which, in my opinion, has improved.
I do not have any further comment.
Reviewer 2 Report
Answers to the doubts raised to the authors
Abstract: “In the abstract there were two sentences that could cause confusión”. The authors have rewritten these statements and now there ins´t confusión. The question is resolved
Introduction: “It was indicated in the introduction, as well as in other sections, that great importance was given to the role of iron absorption inhibition by flavonoids, but this is not the objective of this work”. The authors have modified this point throughout the work and introduced a limitation in which it is considered that the inhibitory effect on iron absorption by flavonoids cannot be demonstrated due to the lack of data on plasma iron. The question is resolved.
Material and Methods: “all the information on flavonoid content comes from a single 24-hour recall. I think that, although this type of questionnaire is validated, a frequency questionnaire and especially a 72-hour recall, including a weekend day, would have provided more complete information on the patients' diet.” The authors point out that there was a mistake. The first revision indicated that the nutritional information comes from only a 24-hour dietary recall, although two questionnaires were actually used: one at the beginning and another a few days later. This has been pointed out in the new revision, and an addition has been made to the limitations section stating that it would have been more appropriate or would have provided more accurate information if a 48- or 72-hour questionnaire had been used. The question is resolved.
“ it would have been interesting to see other antioxidants, such as vitamin E, C or even polyphenols”. The authors have added vitamin E and C as covariates but these two covariates didn’t influence the results. They also update the data in Table 2. The question is resolved.
Results: “In this section there were several issues to resolve, both in format and content.” All questions have been correctly answered and modified by the authors, and this section is now more readable. The issues have been resolved.
Discussion: “The authors are again reminded of the importance given to the inhibitory effect of iron absorption by polyphenols”. The authors adequately justify why this information is included and how it has been modified in the text. The question is resolved
“It does not explain why the relationships observed for Flavan-3-ols disappear when considering the covariates, but not those observed for total flavonoids.” The question is resolved.
“The way of introducing the effect on iron metabolism together, which could be considered as an inducer of anemia, with this risk-reducing role, creates confusion in the reading.” The question is resolved.
There are no comments in this section